# Diversity and distribution of extra-floral nectaries in the cerrado savanna vegetation of Brazil

John Boudouris and Simon A. Queenborough

Department of Evolution, Ecology & Organismal Biology, The Ohio State University, Columbus, OH, USA

## ABSTRACT

**Aim.** Throughout evolutionary history, plants and animals have evolved alongside one another. This is especially apparent when considering mutualistic relationships such as between plants with extra-floral nectaries (EFNs, glands on leaves or stems that secrete nectar) and the ants that visit them. Ants are attracted by the nectar and then protect the plant against destructive herbivores. The distribution of these plants is of particular interest, because it can provide insights into the evolutionary history of this unique trait and the plants that possess it. In this study, we investigated factors driving the distribution of woody plants with EFNs in the cerrado vegetation of Brazil.

**Location.** Brazil

**Methods.** We used a database detailing the incidence of 849 plant species at 367 cerrado sites throughout Brazil. We determined which species possessed EFNs and mapped their distributions. We tested for correlations between the proportion of EFN species at each site and (i) three environmental variables (mean annual temperature, mean annual precipitation, and the precipitation in the driest quarter of the year), (ii) a broad soil classification, and (iii) the total species diversity of each site.

**Results.** We found a wide range in the proportion of EFN species at any one site (0–57%). However, whilst low diversity sites had wide variation in the number of EFN species, high diversity sites all had few EFN species. The proportion of EFN species was positively correlated with absolute latitude and negatively correlated with longitude. When accounting for total species diversity, the proportion of EFN species per site was negatively correlated with precipitation in the driest quarter of the year and positively correlated with temperature range.

**Main Conclusions.** These results suggest either that herbivore pressure may be lower in drier sites, or that ants are not as dominant in these locations, or that plant lineages at these sites were unable to evolve EFNs.

Corresponding author
Simon A. Queenborough,
queenborough.1@osu.edu

## INTRODUCTION

The large-scale distribution patterns of many plant traits, as well as the factors that drive these distributions, are still poorly understood in many ecosystems. In particular, plants

from the tropics were long believed to have more traits associated with defence against herbivores than plants in temperate latitudes (*Schemske, Mittelbach & Cornell, 2009*). The idea that plants in the tropics experience greater herbivore pressure, thus driving selection for improved defence or resistance traits, is key to our understanding of the factors influencing latitudinal gradients in plant traits and patterns of plant diversity.

The geographic implications of the distribution of plant herbivore defenses is complex, in part because there are many variables associated with geographic distribution. Various studies have demonstrated a general trend of increasing anti-herbivore defences in plants moving toward the equator (*Coley & Barone, 1996*; *Fiala & Linsenmair, 1995*; *Oliveira & Oliveira-Filho, 1991*; *Pemberton, 1998*; *Schemske, Mittelbach & Cornell, 2009*). This trend is generally believed to correlate with the intensity of current herbivory in the region, or the intensity of herbivory with which the flora's ancestors had to contend. For example, the presence of cyanide as a defensive compound has been demonstrated to vary in this way with latitude as an effect of temperature gradients (*Jones, Keymer & Ellis, 1978*). Further, the mutualistic relationship between ants (and less commonly, certain wasps and flies) and plants, involving extra-floral nectaries has been demonstrated to increase in abundance with decreasing latitude (*Pemberton, 1998*). Nevertheless, the results of both a recent meta-analysis and a global observational study contradict these findings, suggesting that plants at higher (temperate) latitudes possess greater herbivory resistance traits than plants at lower (tropical) latitudes (*Moles et al., 2011a*; *Moles et al., 2011b*).

Extra-floral nectaries (EFNs) are nectar-producing glands that occur on plant leaves or stems, and are therefore usually not involved in pollination (*Fiala & Linsenmair, 1995*). They are integral components in a particularly interesting mutualistic relationship between plants and insects, most often ants. These glands produce a sugary nectar that attracts ants, which either become resident or frequent visitors of the plant. The presence of ants has been demonstrated to result in lower levels of herbivory for the plant (*Oliveira & Fraitas, 2004*). This benefit is likely the result of the tendency of ant colonies to defend their homes and food sources from possible threats. However, the factors that drive variation in the incidence and abundance of EFN within and among communities remain unknown.

Several studies have determined that EFNs become more prevalent with decreasing latitude (*Fiala & Linsenmair, 1995*; *Keeler, 1980*; *Oliveira & Leitão-Filho, 1987*; *Oliveira & Oliveira-Filho, 1991*; *Pemberton, 1998*), but this information alone is not enough to deduce the cause, because the many potential explanatory variables confound each other in any given ecosystem and they are not always consistent along latitudinal gradients. Further, many such studies compare widely different ecosystems with contrasting evolutionary histories and distantly-related species. For example, the only previous large-scale study to date of EFN prevalence (*Pemberton, 1998*) was conducted in eastern Asia across a number of different biomes, so it is likely that many confounding factors were missed. A more rigorous approach would compare the same life forms and related species in the same ecosystem across a wide range of latitudes and varying environmental conditions. The Brazilian cerrado is a good candidate for this kind of study. The cerrado is a savanna biome that previously covered 2 million km$^2$ (about 22%) of Brazil, with a high level of

biodiversity (*Oliveira & Marquis, 2002*) and a relatively high abundance of plant species with EFN compared to other parts of the world (*Oliveira & Fraitas, 2004*). The cerrado biome covers a wide latitudinal range from 23°S to a few sites close to the equator (*Oliveira & Marquis, 2002*), and is therefore a good place to study latitudinal variation whilst controlling for as many biogeographic and ecological variables as possible.

For this study, we investigated the environmental factors influencing the distribution of EFN. Specifically we ask (i) how the abundance of EFN varied with latitude, precipitation, temperature, soil type, and the species richness?, and (ii) do these relationships support previous established relationships between plant defences and latitude? Based on the current literature we predicted that EFN incidence increased with decreasing latitude (i.e., closer to the equator), and that factors implicated in higher herbivore pressure such as aseasonality would correlate with high EFN incidence.

## MATERIALS AND METHODS

We restricted our analyses to woody plants. To consistently document woody plant incidence across a wide range of latitude and environmental variation, we used the - revised - dataset collected by *Ratter, Bridgewater & Ribeiro (2003)*, revised *Ratter et al. (2010)*, documenting the incidence of 849 tree or large shrub species at 367 cerrado and Amazonian savanna sites throughout Brazil. This dataset currently covers about 75% of the cerrado domain. Other woody communities of the biome such as gallery and mesophytic forests were not included. Species richness varied from Amazonian savannas containing only a single woody species to >100 in the cerrado core area and its southern outliers. All taxa were identified to species-level, making this dataset extremely valuable and virtually unique in large-scale plant datasets (Fig. 1).

Data on which of these species have extra-floral nectaries was obtained from Neotropical floras (*Oliveira & Marquis, 2002*; *Pennington & Ratter, 2006*), an online database (*Keeler, 2008*), and several journal articles (*Oliveira & Leitão-Filho, 1987*; *Oliveira & Oliveira-Filho, 1991*; *Díaz-Castelazo et al., 2004*; *Díaz-Castelazo et al., 2005*; *Oliveira & Fraitas, 2004*; *Marazzi et al., 2006*; *Machado et al., 2008*; *Goitía & Klaus, 2009*; *Schoereder et al., 2010*, see Table S1). Currently, 1–2% of plant species have been confirmed to possess EFN and it is estimated that a further 1–2% of species remain to be discovered with EFN (*Weber & Keeler, 2013*). Thus, our estimates of EFN incidence in the cerrado biome are likely to be conservative, despite the above average knowledge of the species present in the system.

Bioclimatic data were derived from a 30″ gridded dataset consisting of interpolated 50-year normals from New World weather stations (*Hijmans et al., 2005*). Soil data were derived from a digitised version of the 0.0083 (nominally 1-km) resolution Mapa de Solos do Brasil (*EMBRAPA, 1981*), downloaded from the University of New Hampshire, EOS-WEBSTER Earth Science Information Partner (http://eos-webster.sr.unh.edu/home.jsp).

We examined correlates of EFN richness using a generalized linear modelling approach. We modelled the proportion of EFN species at each site as a function of annual mean temperature, temperature range, and temperature seasonality (SD of temperature), annual mean precipitation, precipitation in the driest quarter, and precipitation seasonality

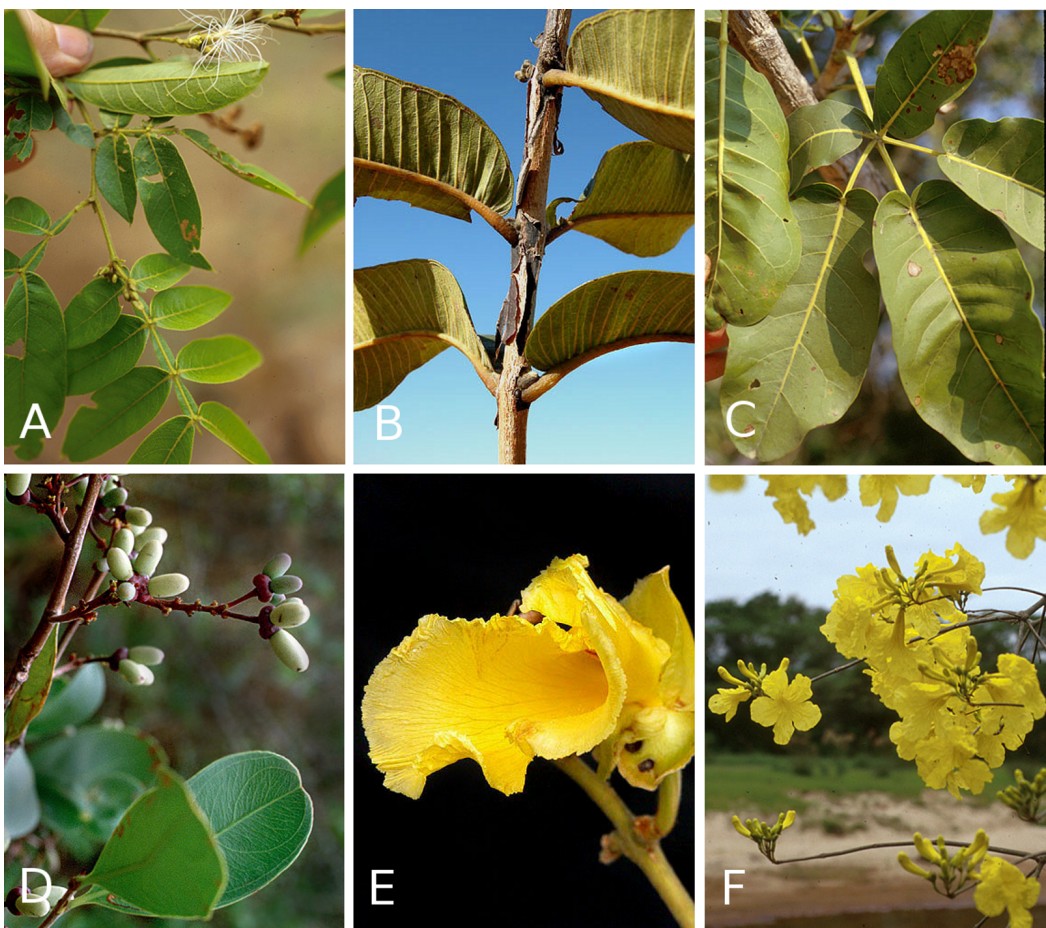

**Figure 1** **Woody plant species with extra-floral nectaries from the cerrados of Brazil.** (A) *Inga vera*, Fabaceae; (B) *Qualea grandiflora*, Vochysiaceae; (C) *Tabebuia aurea*, Bignoniaceae, (D) *Ouratea hexaperma*, Ochnaceae; (E) *Qualea grandiflora*, Vochysiaceae; (F) *Tabebuia aurea*, Bignoniaceae. Credits: Robin Foster, The Field Museum, Chicago, USA (A, C, F); Julio Lombardi, Departamento de Botânica, Instituto de Biociências de Rio Claro, Universidade Estadual Paulista, SP, Brazil (D); Gustavo Schimizu, Dept. Plant Biology, Institute of Biology/Unicamp, Campinas, SP, Brazil (B, E).

(coefficient of variation in precipitation), a broad soil classification, and total species richness. Because we expressed proportion EFN as the number of species with EFNs out of the total species richness per site, we used a binomial error structure. All explanatory variables were rescaled by subtracting the mean value and dividing by the standard deviation to permit comparisons among them.

Finally we tested whether cerrado species with EFN were more widespread than species without EFN (a measure of the ecological success of the species). We modelled the number of occupied sites as a function of EFN incidence, using a generalised linear model with a Poisson error distribution.

## RESULTS

A total of 98 tree and shrub species of the cerrado and Amazonian savannas were documented to have EFN, out of a total of 849 species (Fig. 1). These were distributed

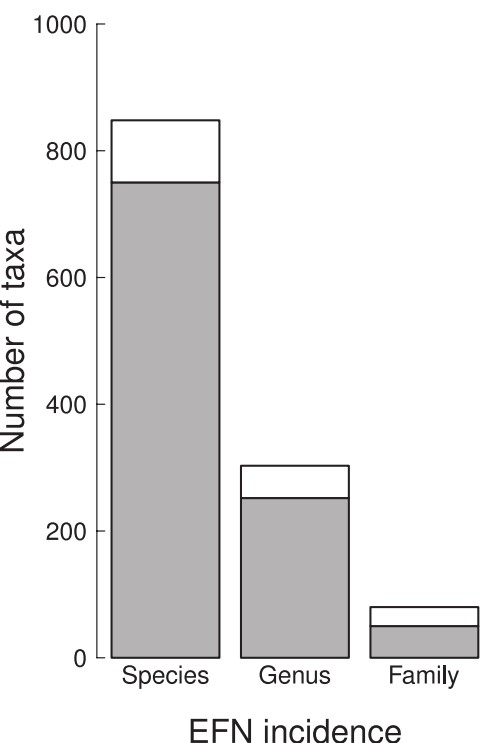

**Figure 2** The number of species, genera and families with (white) and without (grey) extra-floral nectaries in the cerrado of Brazil from a sample of 849 species found in 367 sites.

among 51 genera and 30 families (Fig. 2). The families with the most species with EFN were the Fabaceae, with 31 species, followed by the Bignoniaceae (8 species), Simaroubaceae (6), Malpighiaceae (6), and Chrysobalanaceae (6) each.

Total species richness per site varied from 1 to 212 (mean = 57, sd = 33), and the proportion of species with EFN at each site ranged from 0 to 58% (mean = 22%, sd = 9%, Fig. 3B). There were 39 sites without any species with EFN, and 153 sites with at least 25% of species with EFN. There was a greater proportion of species with EFN in the south and east of Brazil (Figs. 3A, 3C and 3D).

In the full generalised linear model containing all bioclimatic variables, four variables had a statistically significant effect on EFN proportion (Fig. 4, $P < 0.05$). Total species richness, precipitation seasonality, and precipitation in the driest quarter all had negative effects on EFN proportion; temperature range had a small positive effect on EFN proportion (Figs. 3 and 5). These results indicate that cerrado sites with fewer species, lower rainfall in the driest quarter of the year and low seasonality of precipitation all had a higher proportion of species with EFNs. Sites with a greater range of temperature had a slightly greater EFN proportion. There were no statistically significant effect of soil category on EFN proportion.

Of all species in the dataset, 235 species (about 28%) occurred in only one site, 556 species (66%) occurred in <10, and 778 species (92%) occurred in <100 sites. Species

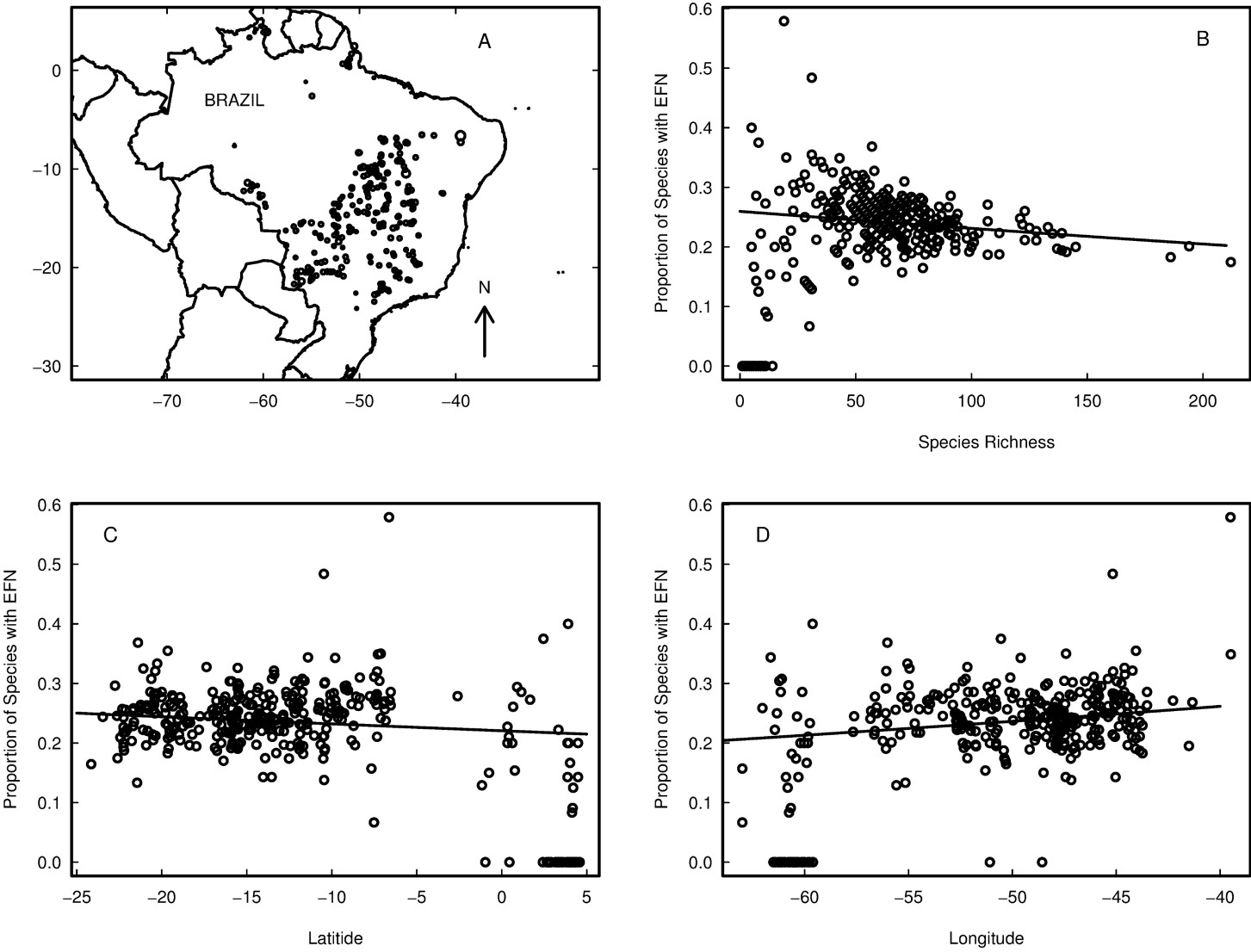

**Figure 3** **Variation in the proportion of species with extra-floral nectaries (EFN) in the cerrado and Amazonian savannas of Brazil.** (A) Location of sample sites. Size of circle is proportional to the proportion of species with nectaries. (B) Proportion of species with EFN as a function of total species richness. (C) Proportion of species with EFN as a function of latitude. (D) Proportion of species with EFN as a function of longitude.

with EFN were on average slightly more widespread than species without EFN, occupying a mean of four sites as opposed to three (Fig. 6).

## DISCUSSION

We found high variability in the incidence of species with EFN species among sites, ranging from 0 to 58%. Furthermore, we found evidence of a latitudinal gradient in EFN proportion, converse to our prediction. Sites further from the equator tended to have slightly more species with EFN than those close to the equator (Fig. 3C), implying more defences at higher latitudes. However, we found a significant longitudinal gradient as well

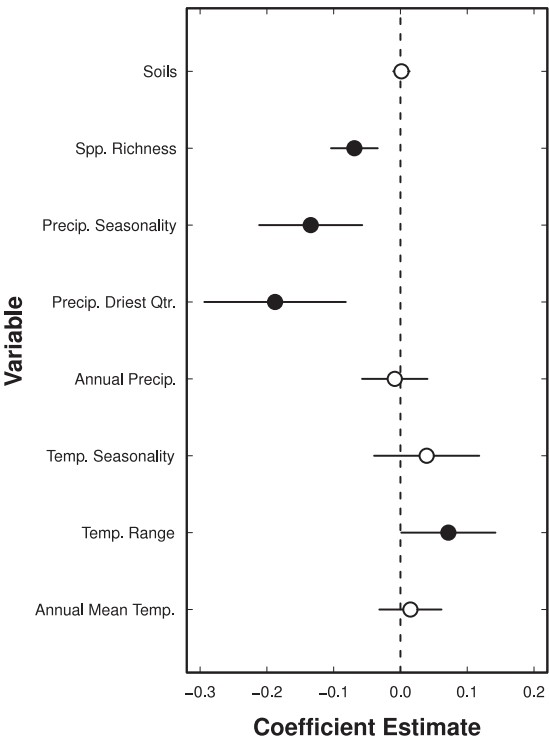

**Figure 4 Coefficient estimates from a binomial generalized linear model of the proportion of species with extra-floral nectaries as a function of standardised bioclimatic variables.** The model also included soil type and total species richness in 367 sites in the cerrado and savanna vegetation of Brazil. Each point indicates the estimate, with thick and thin error bars indicating one and two standard deviations, respectively.

(Fig. 3D), so the situation is more complex than it might appear. What factors cause such variation to occur across a reasonably consistent biome?

Sites with low species richness tended to have a higher proportion of species with EFN. Whilst low richness sites may have inflated proportions EFN because each species contributes more to the overall percentage (i.e., one out of four species is obviously a larger percentage than one out of 20), it is likely that we can consider these results to have some causal implications given the potential advantage of possessing EFN especially in low diversity sites. This is because herbivores are present in any site with vegetation, so the selective pressure for effective herbivore defences like EFNs is always present as well. Further, in low diversity sites, rare species are not 'hidden' by common species, and species-specific herbivores can more easily encounter their particular food source. Unfortunately, no data on the abundance of species at each site are available, so we cannot test whether species with EFN are more abundant than those without.

The trends in the geographic distribution of EFNs show a greater proportion of EFNs as one moves south from the equator and east toward the Atlantic coast. This latitudinal gradient is consistent with the recent findings of *Moles et al. (2011a)*, *Moles et al. (2011b)* and contrary to the idea that herbivory intensity increases toward the equator. Paleoecological evidence suggests that lower latitudes have historically displayed

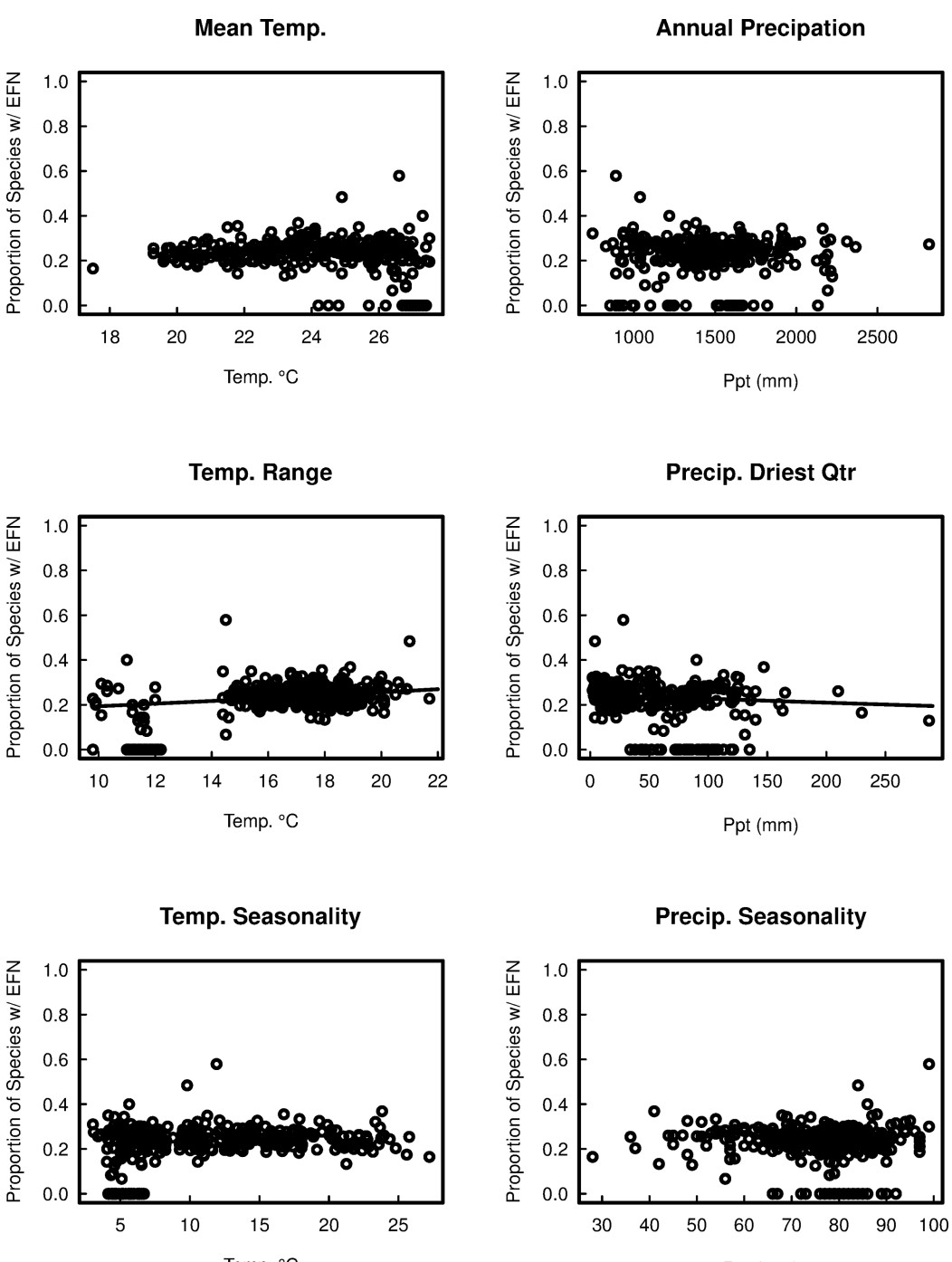

**Figure 5** The proportion of species with extra-floral nectaries in 367 cerrado sites in Brazil as a function of six bioclimatic variables.

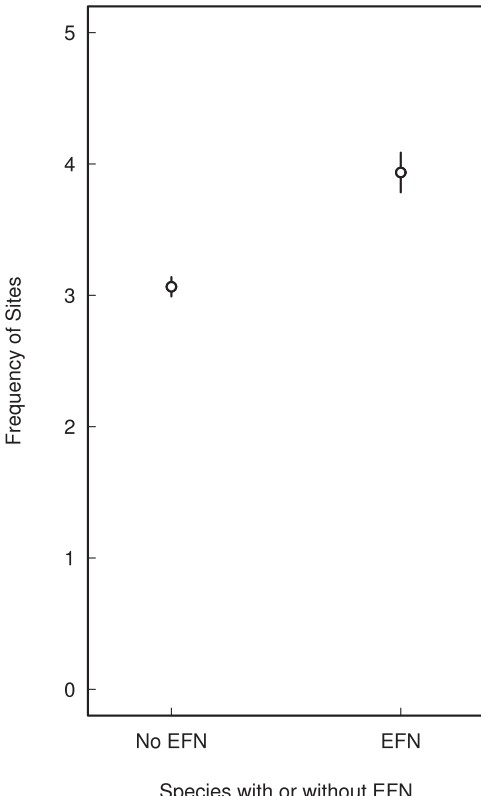

**Figure 6 Range size of woody cerrado species with and without extra-floral nectaries.** Range size was defined as the number of occupied sites out of 367 sites throughout the cerrado and Amazonian savannas of Brazil.

greater herbivore diversity; however, herbivory is not necessarily more intense along this same gradient (*Moles et al., 2011a*). A comprehensive understanding of the latitudinal distribution of herbivore defences would require further inquiry into the latitudinal distribution of herbivory and we cannot necessarily infer implications of the longitudinal gradient from this study. It is possible that the climatic differences of coastal regions may play a role. However, that is not exactly consistent with longitude. There are coastal sites around 55°W and non-coastal sites around 45°W. Further study would need to be devoted to the climatic differences between coastal regions and inland regions.

Climate was significantly correlated with EFN proportion. Clear negative relationships existed between EFN proportion and both precipitation in the driest quarter of the year and precipitation seasonality, but no significant trend was found with total annual precipitation. This suggests that while precipitation is important, it is likely that extremes in precipitation are what drive EFN (or, more likely, ant abundance), rather than total rainfall. This could have something to do with the effects of periods of drought on the intensity of herbivory. One possible explanation, that would need to be investigated further, is that in dry areas, herbivores get a significant portion of their hydration from the vegetation on which they feed.

Sites with a greater annual temperature range had a higher proportion of EFN than sites with a smaller range. Similar to the precipitation results, this implies that it could be the extremes that have the greatest impact on variation. Data on the relationship between the intensity of herbivory and temperature range would be required to fully understand these results.

Finally, in order to fully investigate the relationship between plant defences and herbivore pressure, both sides of the equation must be determined. To date, measuring plant defences has been the more logistically possible. However, effort must be made to estimate herbivory in a consistent manner across ecosystems and biomes in order to fully elucidate the patterns and drivers of variation in plant traits. In terms of traits such as EFN, however, the relationship with insects is also of importance. No data is available on ant abundance across the cerrado, and traits such as EFN may depend more on the availability of the plant's mutualist ants rather than herbivore pressure. Thus, plants in areas of low ant presence or diversity may have evolved rather different defence mechanisms.

In conclusion, we have confirmed *Moles et al.*'s (*2011a*; *2011b*) suggestion that the latitudinal gradient in plant defence traits is more complex than originally thought. At least for one specific defence mechanism (EFN) in the cerrado biome, more species of plants are defended at higher latitudes.

## ACKNOWLEDGEMENTS

We thank James Ratter, Toby Pennington, Sam Bridgewater, and William Milliken for making the cerrado database available, as well as all the numerous fieldworkers who collected the original data. We thank Marjorie Weber for constructive comments on the manuscript.

### Funding

John Boudouris was funded by OSU as an undergraduate research assistant. The funders had no role in study design, data collection and analysis, decision to publish, or preparation of the manuscript.

### Grant Disclosures

The following grant information was disclosed by the authors:
OSU.

### Competing Interests

The authors declare that there are no competing interests.

### Author Contributions

- John Boudouris conceived and designed the experiments, performed the experiments, analyzed the data, wrote the paper.

- Simon A. Queenborough conceived and designed the experiments, performed the experiments, analyzed the data, contributed reagents/materials/analysis tools, wrote the paper.

## Supplemental Information

Supplemental information for this article can be found online at http://dx.doi.org/10.7717/peerj.219.

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
