# Peer review of "Diversity and distribution of extra-floral nectaries in the cerrado savanna vegetation of Brazil"

_PeerJ, doi:10.7717/peerj.219_

## Round 0.1 · original submission · Minor Revisions

Thank you for sending us such an interesting paper. As you can see, both reviewers are strongly supportive our publishing your manuscript. The changes recommended are all minor.

·

Basic reporting

No comments.

Experimental design

No comments.

Validity of the findings

No comments.

Comments for the author

The authors start with an existing data set of 849 plant species at 367 Cerrado sites. They use other data sources to divide the plant species into two groups, those that possess extra-floral nectaries and those that do not. They use yet other data sources to examine correlations between the proportion of extra-floral-nectary-bearing plants, multiple environmental variables, soil type, and total plant species diversity at each site. They report a number of correlations, including an unexpected negative correlation between proportion of extra-floral-nectary-bearing plants and proximity to the equator.

The paper is well-written, the goals and results are clearly stated, and, as far as I can tell, the analyses are well executed. For these reasons I recommend publication. The paper is of general interest for a number of reasons. The Brasilian Cerrado is an important biome deserving of increased study and extra-floral nectaries are important phenomena of broad general interest to ecologists and evolutionary biologists. The correlations revealed by the authors are not easy to interpret with reference to overly general hypotheses, but that is only to be expected from something as complex as the Cerrado, which is actually a mosaic of landscapes with varying geological histories. It's also to be expected from something as complex as extra-floral nectaries, which are the results of many separate convergent evolutionary histories. In spite of these complexities, a set of general evolutionary/ecological mechanisms presumably underlies the origin and maintenance of extrafloral nectaries in the Cerrado. This paper represents a step forward in identifying those mechanisms.

A few minor edits:

P. 4, line 61: Substitute "ask" for "asked."

P. 4, line 65 and elsewhere: "i.e." should be followed by a comma

P. 6, line 104: "The families with [the] most species with EFN [were] the . . ."

P. 7, line 134: "Whilst the low species richness sites may have inflated proportion[s] of EFN . . ."

P. 8, line 142: "Unfortunately, no data . . . [are] available . . ."

P. 9, line 176: "No data [are] available . . ."

Reviewer 2 ·

Basic reporting

This paper investigates the proportion of species bearing extra-floral nectaries in the cerrado of Brazil, and how this relates to potential environmental correlates. The paper was very well written and a joy to read with very few errors (see general comments to orders). It confirms current knowledge in terms of distribution of this trait and suggests further directions of research in relation to environmental factors and herbivory.

Experimental design

The location of the study is appropriate to the question, I don't think you could have picked a better study region. Further it is clear from the acknowledges and references that you have consulted leading taxonomist who have expert knowledge of the region and its flora and therefore the classification of the presence and absence of EFN is likely to be as good as it gets.

I haven't used GLMs in my research but from the little I know the method seems appropriate to the questions being asked.

Validity of the findings

The findings and their interpretation seem to be correct. They back the conclusions of Moles et al. (2011a, b) relating to the distribution and its complexity, and provide a new insight into its relationship with other environmental factors associated with herbivory.

Comments for the author

Minor corrections

Remove the reference from the abstract and there seems to be a / after "0-57" in the abstract that needs deleting

Methods - was it possible to classify all species as having or not having EFN, is there any character uncertainty?

In the results lines 102-106 and 121-124 you mention numbers of species with EFN frequency of sites etc. It may be worth reiterating the total number of species in the flora. i.e. line 102 "A total of 98 tree and shrub species... to have EFN from are total of 367 species." or something similar. I realise you do give this figure in the methods but it is worth reiterating it for comparison with the figures you give in the results.

---

## Round 0.2 · accepted · Accept

Please make the entirely minor changes recommended by the reviews.